# Resolving electron and hole transport properties in semiconductor materials by constant light-induced magneto transport

Artem Musiienko [1] ✉, Fengjiu Yang[1,2], Thomas William Gries [1,3], Chiara Frasca[1,3], Dennis Friedrich [4], Amran Al-Ashouri[1], Elifnaz Sağlamkaya[5], Felix Lang [6], Danny Kojda[7], Yi-Teng Huang [8,9], Valerio Stacchini[1], Robert L. Z. Hoye [9], Mahshid Ahmadi [10], Andrii Kanak [11,12] & Antonio Abate [1,3]

The knowledge of minority and majority charge carrier properties enables controlling the performance of solar cells, transistors, detectors, sensors, and LEDs. Here, we developed the constant light induced magneto transport method which resolves electron and hole mobility, lifetime, diffusion coefficient and length, and quasi-Fermi level splitting. We demonstrate the implication of the constant light induced magneto transport for silicon and metal halide perovskite films. We resolve the transport properties of electrons and holes predicting the material's effectiveness for solar cell application without making the full device. The accessibility of fourteen material parameters paves the way for in-depth exploration of causal mechanisms limiting the efficiency and functionality of material structures. To demonstrate broad applicability, we further characterized twelve materials with drift mobilities spanning from $10^{-3}$ to $10^{3}$ cm$^2$V$^{-1}$s$^{-1}$ and lifetimes varying between $10^{-9}$ and $10^{-3}$ seconds. The universality of our method its potential to advance optoelectronic devices in various technological fields.

Development of the novel semiconducting and semi-insulating materials and improvement of the existing ones rely on the knowledge of the free charge transport properties. In particular, the determination of the minority and majority charge carrier diffusion lengths, lifetimes, mobilities, and concentrations is critical to design and control the effectiveness of semiconductor devices. Currently, it is challenging to probe the minority and majority charge properties due to the limitations of the state-of-the-art experimental methods unable to resolve electron and hole concentration signals. The broadly spread methods such as time-resolved photoluminescence (trPL), terahertz

[1]Solar Energy Division, Helmholtz-Zentrum Berlin für Materialien und Energie GmbH, 12489 Berlin, Germany. [2]Chemistry and Nanoscience Center, National Renewable Energy Laboratory, Golden, CO 80401, USA. [3]Department of Chemistry, University of Bielefeld, Bielefeld, Germany. [4]Institute for Solar Fuels, Helmholtz-Zentrum Berlin für Materialien und Energie GmbH, 14109 Berlin, Germany. [5]Disordered Semiconductor Optoelectronics, Institute of Physics and Astronomy, University of Potsdam, Karl-Liebknecht-Str. 24-25, 14476 Potsdam-Golm, Germany. [6]ROSI Freigeist Juniorgroup, Institute of Physics and Astronomy, University of Potsdam, Karl-Liebknecht-Str. 24-25, 14476 Potsdam-Golm, Germany. [7]Department Dynamics and Transport in Quantum Materials, Helmholtz-Zentrum Berlin für Materialien und Energie GmbH, 14109 Berlin, Germany. [8]Cavendish Laboratory, University of Cambridge, JJ Thomson Ave, Cambridge CB3 0HE, UK. [9]Inorganic Chemistry Laboratory, Department of Chemistry, University of Oxford, South Parks Road, Oxford OX1 3QR, UK. [10]Institute for Advanced Materials and Manufacturing, Department of Materials Science and Engineering, The University of Tennessee Knoxville, Knoxville, TN 37996, USA. [11]Laboratory of Inorganic Chemistry, Department of Chemistry and Applied Biosciences, ETH Zürich, Zürich, Switzerland. [12]Department of General Chemistry and Chemistry of Materials, Yuriy Fedkovych Chernivtsi National University, Chernivtsi 58012, Ukraine. ✉e-mail: artem.musiienko@helmholtz-berlin.de

conductivity, and photoconductivity decay can detect lifetime either for only minority or only majority carriers[1]. In addition, these methods probe charge carriers' properties in a transient regime which is not aligned with device or material operation conditions in a steady state. Thus, the method for accurately characterizing the charge transport properties of electrons and holes (minor and major carriers) is crucial in advancing our understanding of semiconductor materials and optimizing device performance[2]. A detailed comparison of methods capable of material characterization is provided in Tables S1 and S2 in the Supplementary Information (SI).

In the field of material science, researchers have frequently employed the Hall effect as a valuable tool to analyze the behavior of charge carriers and determine their nature, whether they are holes or electrons. The Hall effect, discovered by Edwin Hall in 1879, describes the phenomenon where a material exhibits a response to an applied magnetic field. In Hall's paper[3], they demonstrated the presence of a single carrier type, namely electrons. Until now, the Hall effect has been commonly utilized to determine the concentration and mobility of the majority charge carriers within a material.

In the 20th century, Sakalas et al.[4,5], Höschl et al.[6] and Rosenzweig et al.[7] made significant advancements by demonstrating that a combination of light and the Hall effect could be employed to access electron and hole signals for defect detection in semiconductors. This innovative approach capitalized on the unique characteristic of the Hall effect voltage, which encompasses contributions from both holes and electrons. However, their methodology faced limitations due to the absence of an analytical equation for electron and hole concentrations and the lack of knowledge regarding electron and hole mobility, which impacted the resolution of carrier transport parameters. In recent studies, the Hall effect has also been utilized in combination with light to assess the properties of photocarriers. However, these investigations often operate under the assumption of an equal concentration and lifetimes[8-11] of holes and electrons, thereby

considering a single carrier regime. Consequently, there is a pressing need for improved methods that can accurately characterize the individual properties of both holes and electrons, considering their unequal concentrations, to provide a more comprehensive understanding of electron and hole behavior and transport properties.

In most cases, the carrier concentrations, lifetimes, and diffusion length of electrons and holes are not equal in semiconductors due to the presence of traps capturing minority carriers[2,12-15], as schematically shown in Fig. 1a. To further illustrate the inequality between photogenerated holes ($\Delta p$) and electrons ($\Delta n$), we conducted the simulations of charge generation and recombination in steady-state conditions[9] by using a p-type material with electron traps ($N_t = 10^{14}$ cm$^{-3}$) that have a slightly higher capability for capturing electrons compared to holes. The details of the model can be found in SI §2.3, Eqs. S4–S6. A significant portion of minor electrons become trapped across a wide range of applied light powers because of the activity of electron traps (Fig. 1b). As a result, the concentrations of electrons and holes become equal at high illumination levels when trap saturation occurs. The assumption on equality of concentration of holes and electrons leads to incomplete or sometimes incorrect charge transport description.

In this study, we present the constant light-induced magneto transport (CLIMAT) method, which combines light, electrical current, and a magnetic field (as depicted in Fig. 1c) to assess the transport properties of holes and electrons separately. CLIMAT relies on two key elements: (a) determining electron and hole mobility under two different charge injection regimes, and (b) resolving the electron and hole concentration by utilizing conductivity and Hall coefficient values with developed analytical equations. In addition, correction methods were developed for materials affected by grain boundaries and parasitic conductivity. As a result, CLIMAT resolves electron and hole signals, providing access to 14 parameters of material (mobility ($\mu_e$ and $\mu_h$), concentration ($n$ and $p$), lifetime ($\tau_e$ and $\tau_h$), diffusion length ($L_e$ and $L_h$), diffusion coefficient ($D_e$ and $D_h$), quasi Femi-level splitting (QFLS$_e$,

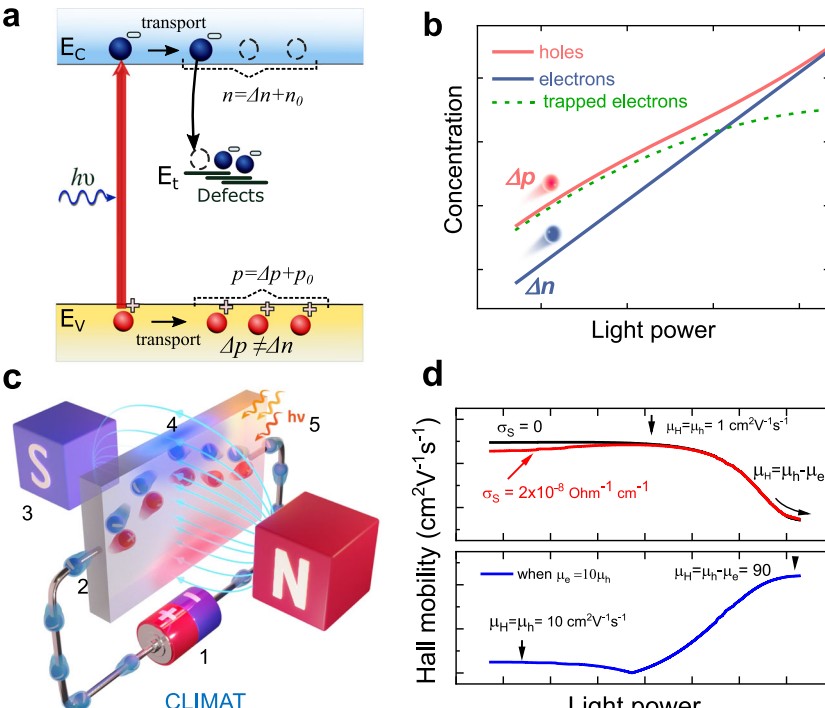

**Fig. 1 | The magnetic field resolves the signal of free carriers generated by light in the CLIMAT method. a** Charge transport model in p-type semiconductor. **b** Simulation of photogenerated holes and electrons in p-type semiconductor. **c** Schematic of CLIMAT method where (1) stands for the current source, (2) for

electrical contact, (3) magnet, (4) induced Hall voltage, and (5) light. **d** Simulation of Hall mobility in p-type material with similar (upper panel) and different (bottom panel) electron and hole mobilities. $\sigma_S$ stands for parasitic conductivity affecting Hall mobility.

QFLS$_h$, and QFLS$_{total}$), and ideality factor (η), in comparison to only two parameters obtained through the classical Hall approach. The application of light significantly extends the range of materials that can be probed by CLIMAT and Hall measurements in general. The free carrier injection and sample conductivity is controlled by the light in CLIMAT thus Hall effect measurements are not limited by the high resistivity or bandgap energy or thin-film thickness. A detailed comparison of CLIMAT, showcasing its superiority over existing state-of-the-art characterization methods, is provided in Tables S1 and S2.

We demonstrated the CLIMAT measurement by applying two photovoltaic materials with notably different charge transport properties and thiknesss: a 500-nm triple-cation metal halide perovskites (MHP) film and a 150-μm silicon wafer sample. Experimental data were employed to fit a simulation model that incorporates charge generation and recombination processes. This enabled us to extract crucial parameters associated with traps responsible for charge losses in the material systems. Moreover, the understanding of charge dynamics provided us with opportunities to identify strategies for reducing charge recombination and enhancing the overall efficiency of semiconductor devices. Furthermore, we have expanded the application of the CLIMAT method to a wide range of thin film and bulk materials, including CdTe, Y6, NaBiS$_2$, MAPbBr$_3$, CsPbBr$_3$, FASnI$_3$, [2-(9H-Carbazol-9-yl)ethyl]phosphonic Acid (2PACz), copper(I)thiocyanate (CuSCN), N,N'-Bis{3-[3-(Dimethylamino)propylamino]propyl}perylene-3,4,9,10-tetracarboxylic diimide (PDINN), and SiC. These materials find diverse applications in radiation sensors[16–20], new generation solar cells[16,21–31], selective materials[16,23,29,32,33], flexible devices[34], light-emitting diodes[12,35,36], memory[37–39], scintillators[40], water-splitting devices[41,42], and transistors[43]. We demonstrated the versatility and potential of the CLIMAT method in characterizing, predicting, and tailoring the properties of these materials, thereby contributing to their advancement and utilization in various technological fields.

## Results

### Constant light-induced magneto transport

In Hall effect measurements probed on conductors or semiconductors, the application of a magnetic field (B) and electrical current (I) causes the deflection of electrons and holes towards one side of the sample, leading to the induction of Hall voltage (V$_H$). In traditional measurements without the light, the majority carriers dominate, thereby providing information solely about conductivity (σ), concentration (n or p), and Hall mobility (μ$_H$). The classical dark Hall effect fails to capture signals from minority carriers and does not provide insights into carrier transport properties, such as electron and hole diffusion length, mobility, and lifetime etc. Here, we present the basis of the CLIMAT method capable of probing electron and hole charge transport properties in the same sample at the same conditions.

CLIMAT combines the measurements of electrical current (I) and longitudinal and transversal voltages (V and V$_H$) with an additional magnetic field and illumination, as depicted in Fig. 1c. In order to distinguish the signals of electrons and holes, we consider their contributions to the experimental conductivity (σ = Id /(S V)) and Hall coefficient (R$_H$ = V$_H$d/(IB)) measured on the semiconductor or semi-insulating sample with the thickness d, which is illuminated by light with generation rate (G). Note that R$_H$ depends on magnetic field B, applied perpendicular to the sample. The light generates both electrons (n) and holes (p), which contribute to both conductivity and Hall coefficient as follows: $\sigma = e(\mu_h p + \mu_e n)$ and $R_H = \frac{(\mu_h^2 p - \mu_e^2 n)}{e(\mu_h p + \mu_e n)^2}$ where $\mu_e$ and $\mu_h$ are the electron and hole drift mobilities, respectively. Upon careful examination of these equations, it becomes evident that the value of σ comprises the sum of hole and electron contributions ($\mu_h p$ and $\mu_e n$) meanwhile R$_H$ represents the difference between hole and

electron signals ($\mu_h^2 p$ and $\mu_e^2 n$). This unique characteristic of the Hall coefficient enables us to separate the signals of electrons and holes. By rearranging the σ and R$_H$ equations, the concentrations of electrons and holes can be directly analytically determined, as demonstrated in Eqs. (1) and (2) (where e is an elementary charge):

$$n = \sigma \times (\mu_h - R_H\sigma) / (e(\mu_h\mu_e + \mu_e\mu_e)) \tag{1}$$

$$p = (\sigma/e - n \cdot \mu_e)/\mu_h \tag{2}$$

To find n and p we need to obtain their respective drift mobilities. This can be achieved by varying the intensity of light and analyzing the Hall mobility. The Hall mobility, which comprises contributions from both electrons and holes (as shown in Eq. (3)), can be calculated using the conductivity and Hall coefficient ($\mu_H = |\sigma \cdot R_H|$). In the following section, we will discuss holes and electrons as major and minor carriers, aiming to generalize the CLIMAT technique for different materials and doping levels. We assume that $\mu_h$ and $\mu_e$ are independent of illumination[44].

$$\mu_H = \left| \frac{(\mu_h^2 p - \mu_e^2 n)}{(\mu_h p + \mu_e n)} \right| \tag{3}$$

$$\begin{cases} p \gg n \implies \mu_H = \mu_h; \\ n \gg p \implies \mu_H = \mu_e; \\ p = n \implies \mu_H = |\mu_h - \mu_e| = \Delta\mu \end{cases} \tag{4}$$

Equation (3) predicts that the Hall mobility will be equal to the mobility of the majority carrier in the absence of light (dark condition), as indicated in Eq. 4. In addition, at high illumination intensities where the carrier concentration surpasses the trap concentration (saturation of trap), the concentrations of holes and electrons tend to merge (n ≈ p). In this case, the Hall mobility is equal to the absolute difference between $\mu_h$ and $\mu_e$, denoted as $\Delta\mu = |\mu_h - \mu_e|$ (Eq. (4)).

It is important to consider the possible sign change of σ • R$_H$ to accurately determine the mobility of the minor carriers. As we demonstrated below, in samples where the minority charge carriers exhibit higher mobility compared to the majority charge carriers such as p-type silicon, the mobility of the minority carriers is determined by adding Δμ and the mobility of the majority carriers found under dark or low illumination conditions. Conversely, if the majority carriers have higher mobility compared to the minority charge carriers, like in p-type halide perovskite, the mobility of the minority carriers is determined by subtracting Δμ from the mobility of the major carriers. To determine which mobility is larger, we utilized the sign of R$_H$. The sign of R$_H$ reveals the carrier type with higher mobility when n = p at high light intensity. The change in the balance of electron and hole contributions, ($\mu_h^2 p$ - $\mu_e^2 n$), provides a clue regarding which mobility is greater. If the minority carriers have higher drift mobility, the change in the balance of electron and hole contributions, by increasing the light illumination results in a change in the sign of R$_H$ (from negative to positive or from positive to negative). Conversely, if the sign of R$_H$ remains unchanged under light illumination, it implies that the majority carriers possess higher mobility relative to the minority charge carriers. By employing this approach, we can identify and determine the electron and hole mobilities. We summarized generalization of this concept in Table S1. Figure 1d illustrates an example simulation of Hall mobility, showcasing the saturation of Hall mobility under high intensities (I) to the value of |$\mu_h$ - $\mu_e$| in materials containing one deep trap and with $\mu_e$ = 10 × $\mu_h$ = 100 cm$^2$V$^{-1}$s$^{-1}$ as well as $\mu_e$ = $\mu_h$ = 1 cm$^2$V$^{-1}$s$^{-1}$.

After finding mobility, concentration values can be found by Eqs. (1) and (2). The charge transport parameters of electrons and

holes such as lifetime ($\tau_e$ and $\tau_h$), diffusion length ($L_e$ and $L_h$), diffusion coefficient ($D_e$ and $D_h$), quasi Femi-level splitting (QFLS$_e$ and QFLS$_h$), and ideality factor ($\eta$) can be calculated as follows: $\tau_e = \frac{\triangle n}{G}$, $\tau_h = \frac{\triangle p}{G}$, $D = \mu k_B T/e$, $L_e = \sqrt{D_e \tau_e}$ $L_h = \sqrt{D_h \tau_h}$, $QFLS_e = kT\ln(\frac{n}{n_i})$, $QFLS_h = kT\ln(\frac{p}{n_i})$, and $\eta = \frac{e(QFLS_2 - QFLS_1)}{\ln(G_2/G_1)k_B T}$. It's worth mentioning that lifetime, $\tau$, of free carriers is defined as ration between photogenerated carrier density and generation rate similar to previous reports[8,11,44]. Steady-state $\tau$ represents time and $L$ represents distance during which mobile charge carriers (electrons or holes) remain in an active, transport-ready state before undergoing recombination or becoming trapped by defects. The value of $\tau$ probed by CLIMAT accounts for both radiative and non-radiative (trap-associated) recombination processes, as well as takes into consideration trap filling (Eq. S7 and S8). The generation rate ($G$) is directly found by using a calibrated photodetector (more information in SI §1.2) . Notably, CLIMAT can probe the Qusai Fermi Level Splitting (QFLS$_{e,h}$) of both electrons and holes by directly measuring the concentration of electrons and holes, which is closely linked to the highest open circuit voltage ($V_{OC}$) of the materials. This implies that CLIMAT can predict the performance of a material as a solar cell absorber without requiring the complete solar cell device.

Note that charge transport parameters $\tau$, $\mu$, $\sigma$, $R_H$, $L$, QFLS, $n$, $p$, and $\eta$ are not constant and change as a function of generation rate, as will be shown later in this study. In our model, we assume drift mobility to be independent[11,44] on the generation rate as confirmed by the saturation of the Hall mobility to the constant value without a further change, particularly an increase, of the Hall mobility upon saturation. For more in-depth technical information regarding the CLIMAT module design and algorithms can be found in SI (§1 Figs. S2–S4, S6 and S7). The proposed methodology (Eqs. (1)–(4)) applies to both p-type and n-type materials. The consideration of completely intrinsic material is given in SI Eqs. S1 and S2.

### Correction of CLIMAT at low signal conditions and parasitic conductivity

Parasitic conductivity ($\sigma_S$), such as grain boundaries, surface, ions, and non-uniformities, has an impact on Hall effect measurements in specific material systems[45] $\sigma_S$ contributes to the overall sample conductance ($\sigma = (\sigma_S + e\mu_h p + e\mu_e n)$) and decrease the Hall effect signal due to the low mobility of parasitic carriers. When considering Hall mobility and conductivity, the value of $\sigma_S$ must be included in the calculations. This can be expressed as follows: $\mu_H = |e(\mu_h^2 p - \mu_e^2 n)|/(\sigma_S + e(\mu_h p + \mu_e n))$. In order to overcome the impact of $\sigma_S$, the conductivity contribution of free carriers in the bulk material can be enhanced through sample illumination. As a result, the contribution of photogenerated holes ($\mu_h p$ in p-type material as an example) surpasses the value of $\sigma_S$, leading to an increase in the overall value up to the level of the actual major carrier drift mobility. This phenomenon occurs in the low illumination regime ($p > n$ or $n > p$).

Once the major carrier mobility is determined as the peak value, the dark conductivity ($\sigma_0$) and $\mu_H$ can be utilized to estimate the correct dark concentration: $p_0 = \mu_H \sigma_0/(e(\mu_h)^2)$. To provide an example of this simulation, let's consider a semiconductor sample with $\sigma_S = 2 \times 10^{-8}$ $\Omega^{-1}$cm$^{-1}$, one deep trap, and $\mu_e = \mu_h = 1$ cm$^2$V$^{-1}$s$^{-1}$, as shown in Fig. 1d (top, red curve). The simulation illustrates the increase in Hall mobility towards the correct free hole mobility value.

In addition, when electrons and holes possess comparable or identical mobility values, the Hall mobility tends to decrease at higher intensities due to the cancellation effect between electrons and holes, as depicted in Fig. 1d (top, black curve). In order to disregard the influence of parasitic conductivity on Hall mobility and determine the mobility of minority carriers, we can rely on neighboring Hall mobility data points obtained as a function of intensity as shown in Eq. (5). Further elaboration and derivations of this concept across various

material types can be found in the SI §2.2 Eq. S3 and Fig. S8.

$$\begin{cases} e(\mu_h p + \mu_e n) \gg \sigma_S & \Rightarrow \quad \mu_h = \max(\mu_H(G)) \\ p \approx n & \Rightarrow \quad |\mu_h - \mu_e| = |(\mu_{H2}\sigma_2 - \mu_{H1}\sigma_1)/(\sigma_2 - \sigma_1)| \end{cases} \quad (5)$$

Some materials—such as MHP and organic semiconductors—demonstrate similar values of the free hole and electron mobility, which lead to a very low Hall effect signal as electrons and holes cancel each other out (Eq. (3)). In such conditions, the missing points (where Hall signal vanishes, typically at high intensities >0.1 Suns) of the electron and hole concentration can be calculated by using conductivity value $n = p = \sigma/(e(\mu_h + \mu_e))$.

### Characterization of p-type silicon with different $\mu_e$ and $\mu_h$ by CLIMAT

The developed CLIMAT method enables the discovery of fourteen crucial material parameters that are not attainable through classical Hall measurements or single-carrier approaches. To illustrate the relevance of this technique, we first apply the CLIMAT method to p-type silicon ($d = 150$ μm), which exhibits a substantial disparity between $\mu_e$ and $\mu_h$. To minimize the cost and facilitate the practical implementation of CLIMAT, a flat LED illumination was employed. The design of the CLIMAT module is presented in Fig. S2. To determine the electron and hole charge transport parameters in the silicon sample, a series of measurements were performed. Firstly, $\sigma$, $R_H$, and $\mu_H$ were measured as a function of generation rate using an 827-nm LED and 4-probe Hall measurements with an AC magnetic field, as illustrated in Fig. 2a–c. In the dark, the silicon sample exhibits a positive Hall effect signal, confirming its p-type conductivity. The dark condition reveals $\mu_h$ of 328 cm$^2$V$^{-1}$s$^{-1}$, with a dominant hole concentration $p_0 = 4.6 \times 10^{14}$ cm$^{-3}$.

As the intensity of illumination increases, the Hall signal (Fig. 2b) undergoes a sign reversal. This reversal is attributed to the well-established fact that electrons have significantly higher mobility in silicon. Due to the substantial impact of electrons ($-\mu_e^2 n$) in the Hall signal, the Hall effect becomes negative when $n$ approaches the value of $p$. At generation rates ($G$) greater than $10^{20}$ cm$^{-3}$s$^{-1}$, we observe an apparent saturation of Hall mobility due to the equal $n$ and $p$ (in agreement with Eqs. (3) and (4)). This saturation point allows us to determine $\Delta\mu = 1240$ cm$^2$V$^{-1}$s$^{-1}$. Considering the sign change in $R_H$, we found $\mu_e = 1568$ cm$^2$V$^{-1}$s$^{-1}$ by adding $\mu_h$ to $\Delta\mu$. Furthermore, $D_e = 40.6$ cm$^2$s$^{-1}$ and $D_h = 8.5$ cm$^2$s$^{-1}$ are assessed using the mobility values. The value of drift mobility is in good agreement with Terahertz spectroscopy, showing a value of 1581 cm$^2$V$^{-1}$s$^{-1}$ (Fig. S12c).

Furthermore, we found $n$ and $p$ individually using Eqs. (1) and (2). Figure 2d illustrates $n$ and $p$ as a function of generation rate up to one sun power. Consistent with expectations for p-type semiconductors, the concentration of holes prevails under low and moderate illumination conditions, up to $G = 10^{20}$ cm$^{-3}$s$^{-1}$, beyond which $n$ and $p$ become equal. By resolving $n$ and $p$ as a function of $G$, we further investigate the charge transport properties of each carrier type. The hole lifetime (Fig. 3a) shows a decrease from $8.5 \times 10^{-7}$ s as $G$ increases, to a value of $1.16 \times 10^{-7}$ s at $G = 1.5 \times 10^{21}$ cm$^{-3}$s$^{-1}$ (one sun illumination). On the other hand, the electron lifetime remains relatively unchanged with increasing $G$, reaching a similar value of $1.14 \times 10^{-7}$ s under one sun. This is typical evidence for p-type semiconductors, where $\tau_h$ is influenced by the density of minority carriers (i.e., electrons). It is important to note that $n$ and $p$ as well as $\tau_e$ and $\tau_h$ become similar only at moderate $G$ exceeding $10^{20}$ cm$^{-3}$s$^{-1}$. The carrier lifetime value aligns well with the results from time-resolved microwave conductivity(TRMC)[1], indicating a measurement of $1.7 \times 10^{-7}$ s (Fig. S12a).

Using the obtained values of lifetimes and drift mobilities, we calculated $L_e$ and $L_h$ as depicted in Fig. 3b. These $L_e$ and $L_h$ mimic the dependence of lifetime on $G$ and reach values of 21.2 and 9.7 μm, respectively, under one sun. The observed values of the charge

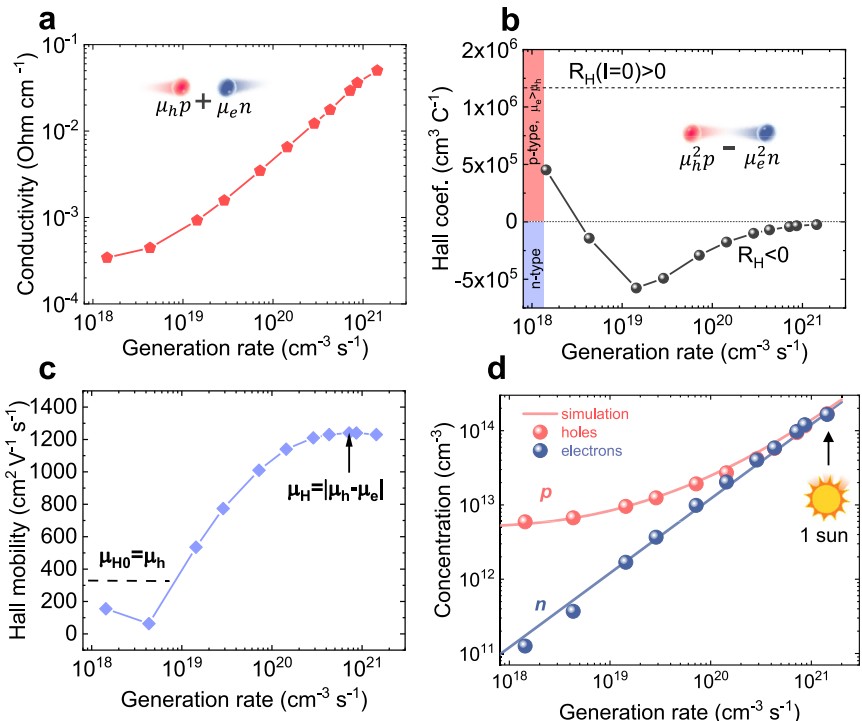

**Fig. 2 | Resolved electron and hole signals by constant light-induced magneto transport technique in *p*-type silicon. a** Conductivity, **b** Hall coefficient, and **c** Hall mobility were measured by the 4-probes CLIMAT as a function of photon generation rate. **d** Electrons $n(G)$ and holes $p(G)$ concentrations as a function of the generation rate as extracted by CLIMAT.

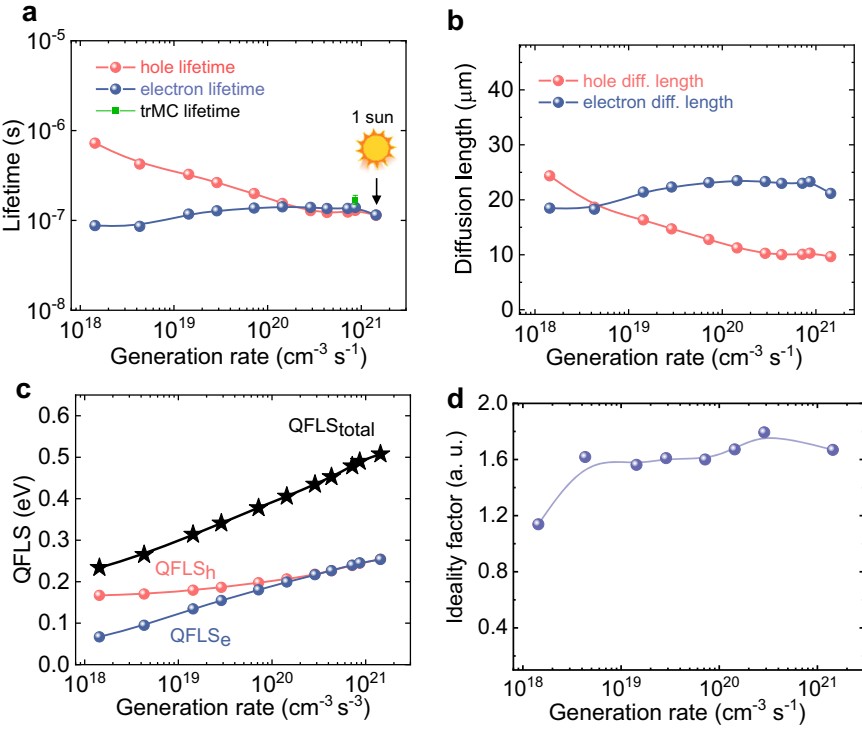

**Fig. 3 | Electron and hole properties in *p*-type silicon resolved by CLIMAT. a** Carrier lifetimes, **b** diffusion lengths, **c** QFLS (found by QFLS$_e$ + QFLS$_h$), and **d** $\eta$ as a function of the generation rate as extracted by CLIMATs.

transport parameters align with previously reported values[46] found in both n-type and p-type silicon individually. By comparing the values of diffusion length with the material's thickness (150 µm > $L$), we enclosed that when this material is used in solar cell applications, free carriers will reach selective contact with significant charge losses, especially free holes.

By directly determining $n$ and $p$, we can make predictions regarding the performance of the investigated silicon sample as an

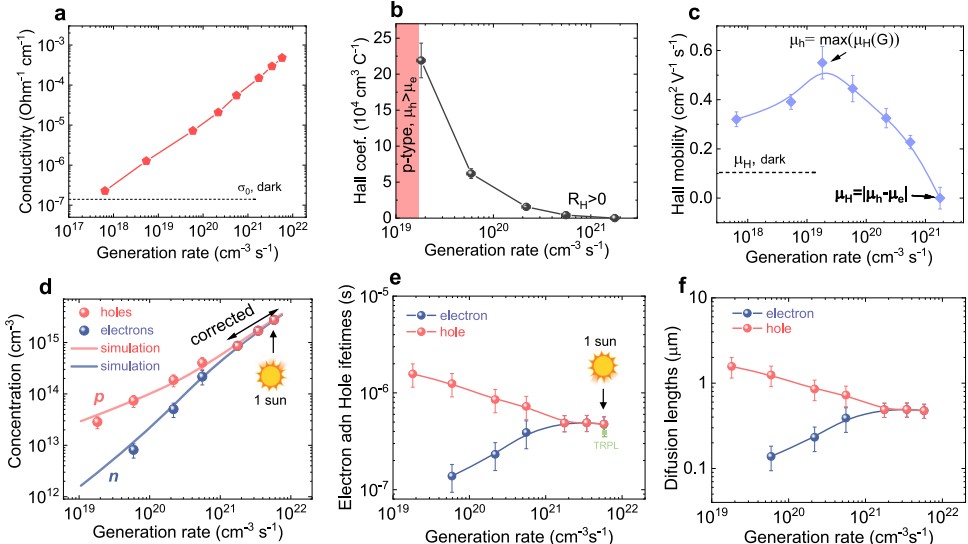

**Fig. 4 | Resolved electron and hole properties by CLIMAT in *p*-type perovskite.** **a** Conductivity, **b** Hall coefficient, and **c** Hall mobility are measured by the 4-probes CLIMAT as a function of photon generation rate. **d** Electrons $n(G)$ and holes $p(G)$ concentrations as a function of $G$ as extracted by CLIMAT. Solid red and blue lines show simulation. **e** Electron and hole lifetimes as a function of $G$. **f** Carriers' diffusion lengths as a function of $G$.

active material in solar cell devices. The silicon samples exhibit QFLS of 0.56 eV under 1 sun illumination, Fig. 3c. To establish a reliable comparison, we also assessed QFLS using a different method, photoluminescence quantum yield (PLQY), which yielded a value of 0.58 eV under the same conditions, Fig. S14. This comparison demonstrates the consistency and reliability of the CLIMAT method. Importantly, tracking individual $L_e$, $L_h$, QFLS$_e$ and QFLS$_h$ allows for a better understanding of the electron and hole limitations in silicon.

Furthermore, we calculated ideality factor $\eta_{Si} = 1.6$, which provides insights into the dominant charge recombination mechanism. Ideality factors falling within the range of $1 < \eta < 2$ indicate the influence of non-radiative recombination (Shockley–Read–Hall) on charge transport[47]. On the other hand, $\eta$ does not provide information on the properties of traps responsible for charge carriers recombination. By acquiring knowledge of $n$ and $p$ as a function of $G$, we can construct charge transport simulations using a theoretical model that incorporates non-radiative and radiative recombination (Eqs. S4–S6). Conducting these simulations in a steady-state condition we fitted the experimental results (Fig. 2d blue and red lines) and found the trap parameters: electron capture cross-section $\widetilde{\sigma}_e = 0.3 \cdot 10^{-15}$ cm$^2$, hole capture crossection $\widetilde{\sigma}_h = 4 \cdot 10^{-14}$ cm$^2$, concentration $N_t = 3 \cdot 10^{15}$ cm$^{-3}$, and activation energy $E_t = 0.3$–$0.6$ eV. The main uncertainty of $E_t$ lies in the fact that the trap remains unoccupied in a wide range of energies above the Fermi level.

## Characterization of p-type metal halide perovskite semiconductor with similar $\mu_e$ and $\mu_h$ by CLIMAT

Here, we showcase the characterization of an MHP material (0.9FACsPbI$_2$Br + 0.1MAPbCl$_3$) where the mobilities of holes and electrons are similar. It is important to note that the perovskite film is much thinner (500 nm) compared to the silicon sample (150 μm), which means that the parasitic conductivity of surface and grain boundaries can influence the Hall effect data. To address this issue, we applied a correction procedure developed in this study (Eq. (5) and Fig. S8) to accurately determine the charge transport parameters in perovskite films.

The MHP sample exhibits a dark hole concentration ($\widetilde{p_0}$) of $10^{13}$ cm$^{-3}$ and a Hall mobility of 0.08 cm$^2$V$^{-1}$s$^{-1}$, as determined using the classical Hall approach. The conductivity, Hall coefficient, and $\mu_H$ data as a function of $G$ are presented in Fig. 4a–c. To generate free carriers in the perovskite film, we utilized a 617-nm LED as the illumination source. We observe an increase in σ and a decrease in the $R_H$ attributed to the corresponding increase in carrier density. Initially, the $\mu_H$ shows an upward trend, reaching a maximum at $G = 3 \times 10^{19}$ cm$^{-3}$s$^{-1}$, after which it begins to decline. This initial increase in $\mu_H$ indicates the influence of parasitic conductivity on the Hall data, as predicted by Eq. (5). The subsequent decrease in $\mu_H$ can be attributed to the comparable values of electron and hole mobility compensating each other due to their opposite charge polarities (Eq. (3), Fig. 1d (top)).

By applying the correction approach, we identify a peak value of 0.55 cm$^2$V$^{-1}$s$^{-1}$ for the mobility of photogenerated holes in Fig. 4c. This peak corresponds to the point where the conductivity of photogenerated holes surpasses the contribution from σ$_S$. Consequently, we can correct the data obtained from the dark Hall measurements and unveil the accurate values of $p_0 = 6 \times 10^{10}$ cm$^{-3}$ and σ$_S = 8 \times 10^{-8}$ Ω$^{-1}$cm$^{-1}$. It is worth noting that the corrected value of the hole concentration is more than two orders of magnitude lower than the value obtained through the classical approach. This stark contrast highlights the superior capabilities of the CLIMAT correction approach. We have also determined the difference between the hole and electron mobility to be $\mu_h - \mu_e = 0.03$ cm$^2$V$^{-1}$s$^{-1}$ by utilizing Eq. (5) and the neighboring data points of $\mu_H$ and σ (at $G = 10^{21}$ cm$^{-3}$s$^{-1}$). The obtained value of electron mobility, 0.52 cm$^2$V$^{-1}$s$^{-1}$, aligns perfectly with our initial hypothesis of similar mobility values for holes and electrons. Furthermore, we calculated $D_e = 1.35 \times 10^{-2}$ cm$^2$s$^{-1}$ and $D_h = 1.42 \times 10^{-2}$ cm$^2$s$^{-1}$. The order of magnitude of σ$_S$ is consistent with previously reported values ($10^{-8}$ Ω$^{-1}$cm$^{-1}$) in halide perovskite semiconductors[45]. σ$_S$ has negligible effect on charge transport at generation rates larger than $10^{19}$ cm$^{-3}$s$^{-1}$.

With knowledge of $\mu_h$ and $\mu_e$, we were able to determine the values of $n$ and $p$ using Eqs. (1) and (2). To ensure accurate carrier concentration values, we employed a low signal correction algorithm based on conductivity data (Fig. 4a) for points where the Hall voltage dropped to zero (the last three points in Fig. 4b, d). The concentration analysis revealed a predominance of free holes up to a generation rate of $10^{21}$ cm$^{-3}$. Beyond this point, the concentration of electrons and holes began to merge, as depicted in Fig. 4d. The merging of $n$ and $p$ is consistent with the observed decrease in Hall mobility below the resolution of our setup (less than 0.01 cm$^2$V$^{-1}$s$^{-1}$ in this sample). The lifetimes and diffusion lengths of electrons and holes are calculated based on $n$ and $p$ and shown in Fig. 4e, f. As expected in a p-type

material, the lifetime of holes decreases with an increase in the generation rate, ranging from 1.6 μs to 476 ns. Interestingly, the electron lifetime exhibits an increase (from 138 to 476 ns), which can be attributed to the occupation of electron traps. To compare the determined lifetimes of carriers with other methods, we conducted trPL measurements at an equivalent fluence of 1 sun. The lifetime obtained from CLIMAT aligns well with the decay constant of 400 ns observed in trPL measurements, as shown in Fig. S12. It is important to note that trPL is unable to resolve the lifetimes of electrons and holes at lower intensities, where CLIMAT detects a distinct difference up to one order of magnitude.

The diffusion length, determined from the values of lifetime and mobility, exhibits a similar dependence on the generation rate. Under one sun illumination power, the electrons and holes demonstrated comparable diffusion lengths, with $L_e \approx L_h = 0.48\,\mu m$. As we found, MHP have balanced charge transport under high illumination compared to silicon. $L_e$ and $L_h$ are similar to the thickness of the material (0.5 μm), pointing to the effective transport of free charges in such a thin film. The steady-state diffusion lengths of the electrons and holes are in good agreement with previously reported values in literature[8,11,48]. Furthermore, we estimated the total QFLS$_{pero}$ to be 1.43 eV at 1 sun, which is close to 1.38 eV probed by the PLQY. The ideality factor ($\eta_{pero}$) of 1.8 indicates non-radiative recombination losses. We performed a charge recombination simulation to determine traps parameters to be, $\tilde{\sigma}_e = 2 \cdot 10^{-15}\,cm^2$, $\tilde{\sigma}_h = 8 \cdot 10^{-16}\,cm^2$, $N_t = 10^{14}\,cm^{-3}$, and $E_t = 0.4$–$1.2$ eV.

In our CLIMAT approach, we assume that the true drift mobilities ($\mu_e$ and $\mu_h$) remain unaffected by illumination. This assumption is based on the fact that the carrier density resulting from the light injection is lower than $10^{17}\,cm^{-3}$, a threshold beyond which the drift mobility can be influenced by possible charged defect scattering[49,50]. In addition, our experimental observations do not provide any evidence of Hall mobility variations after saturation, as shown in Fig. 2c or Fig. 4c for high intensities. The impact of grain boundaries is included in our approach (Fig. S8 and Eq. (5)).

To further demonstrate the universality of CLIMAT for the diverse material science community, we applied CLIMAT to various semiconductor materials. We focused on the mobility and lifetime of charge carriers as shown in Fig. 5. The observed ranges of mobility spanned from $10^{-3}\,cm^2V^{-1}s^{-1}$ in organic PDINN to $10^3\,cm^2V^{-1}s^{-1}$ in silicon, while lifetimes ranged from 1 ns in SiC to 1 millisecond in sc-CsPbBr$_3$. We also included properties of single-crystal of MAPbBr$_3$ and CsPbBr$_3$ materials, both probed in the low signal regime ($p$, $n < 10^{13}\,cm^{-3}$). This allows us to compare CLIMAT data with Time of Flight (ToF) data measured for the same materials (Table S3). The results revealed a good agreement between the CLIMAT and ToF data, which further validated the universality and effectiveness of our method in accurately characterizing these optoelectronic materials. In addition, we applied CLIMAT with a single carrier algorithm (Fig. S5) to study highly doped charge transport layers commonly used in solar cells, including FASnI$_3$, CuSCN, 2PACz, and PDINN. Since these systems exhibit negligible minority carrier signals due to their extremely low minority carrier lifetimes, our primary emphasis is on determining the majority carrier lifetime and mobility. These parameters play a critical role in optimizing charge extraction layers for solar cell devices and LEDs, where the majority carriers govern the charge transport.

## Discussion

Detecting the properties of minority and majority charge carriers in semiconductor and semi-insulating materials presents a challenging task. This lack of knowledge hinders the development of materials for a wide range of optoelectronic devices. To address this limitation, we have developed CLIMAT method, as a powerful tool for characterizing charge transport properties in semiconductor materials. By combining

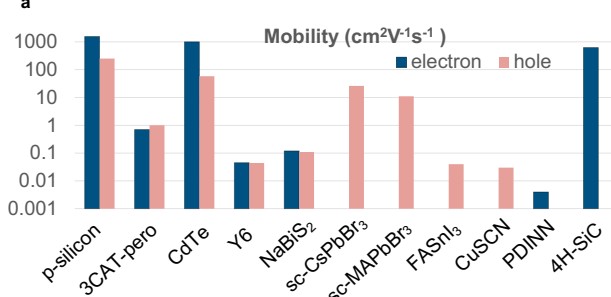

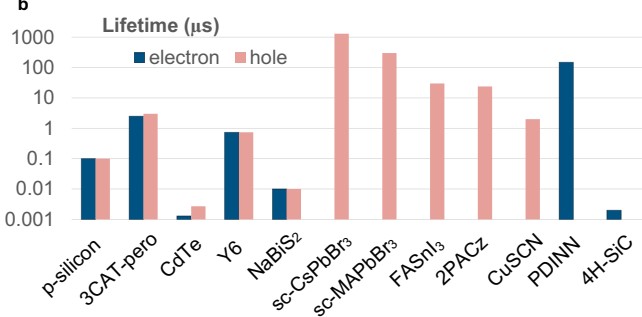

**Fig. 5 | Charge transport in various optoelectronic materials probed by CLIMAT. a** Mobility and **b** Lifetime. Single-crystal (sc)-CsPbBr3, sc-MAPbBr3, and 4H-SiC were characterized at low injection conditions ($n < 10^{13}\,cm^{-3}$ and $p < 10^{13}\,cm^{-3}$) to compare mobility and lifetime of majority carries with a time of flight data Fig. S13 and Table S3. FASnI3, CuSCN, and PDINN are highly doped HTLs and ETLs with carrier densities larger than $10^{15}\,cm^{-3}$[51,52]. In such a system charge transport is dominated only by the majority carriers; thus, minority carriers' mobility and lifetime are not demonstrated. The literature value of hole mobility ($10^{-4}\,cm^2V^{-1}s^{-1}$) was used to calculate the lifetime of free holes in 2PACz[53] by the correction algorithm in Fig. S9. Note that for organic semiconductors, CLIMAT probes properties of carriers in the charge transport state[54].

light, electrical current, and a magnetic field, CLIMAT separately assesses the transport properties of electrons and holes, providing valuable insights into the individual behavior of these carriers. Compared to classical Hall measurements, which probe only two parameters (majority carrier concentration and mobility), CLIMAT can determine fourteen material properties, including $n$, $p$, $\mu_e$, $\mu_h$, $\tau_e$, $\tau_h$, $L_e$ $L_h$, $D_e$, $D_h$, QFLS$_e$, QFLS$_h$, QFLS, and $\eta$, as a function of illumination intensity or carrier density.

We demonstrated the capability of CLIMAT to characterize two different materials with substantial differences in electron and hole properties – silicon and MHP. Contrary to previous studies assuming $\tau_e = \tau_h$, our results demonstrated up to one order of magnitude differences between the values of electron and hole concentrations and lifetimes, which only became similar at high illuminations closer to 1 sun. To overcome the limitation of the CLIMAT setup for perovskite materials (low Hall signal), we applied a correction method that accurately determines carrier mobility and concentration, as well as parasitic conductivity of grain boundaries. By knowing $n$ and $p$ as a function of generation rate, we determined recombination mechanisms and parameters of traps. Our study represents the pioneering achievement of simultaneously determining all characteristics of minority and majority carriers from a single experimental setup, applied to a single sample, as compared to other methods in Tables S1 and S2. The charge transport properties obtained from CLIMAT were in perfect agreement with results from PLQY, trPL, and TOF techniques.

We demonstrated that CLIMAT enables the determination of electron and hole charge transport properties at various controlled

carrier injection level ranges, allowing for the mimicry of charge transport in specific device configurations. This capability can be applied to probe materials in low illumination regimes in sensors or high illumination regimes in solar cells, transistors, and LEDs. We characterized materials finding various applications in optoelectronic devices. The materials exhibited a wide range of drift mobilities, spanning from $10^{-3}$ to $10^{3}$ cm$^2$V$^{-1}$s$^{-1}$, and lifetimes varying between $10^{-9}$ and $10^{-3}$ s demonstrating the range of CLIMAT's applicability across the broad material science field.

## Methods

### Constant light-induced magneto transport (CLIMAT)

In the context of magneto transport, we refer to the transport of charge carriers (such as free electrons or free holes) in the presence of a magnetic field, which includes phenomena like the Hall effect. The Hall effect is a part of the magneto transport measurements family[55–57]. Hall effect is used to investigate the charge transport properties of materials and as a sensor for measuring the strength of a magnetic field. CLIMAT measurements were performed by using an AC magnetic field with a lock-in amplifier (Lake Shore System). The amplifier is used to improve the Hall effect signal because of the low mobility and conductivity of the metal halide perovskite materials. The samples were characterized by using a four-probes (Van Der Pauw) at room temperature. For the perovskite semiconductor, glass encapsulation was sued to prevent sample degradation due to water in the atmosphere. Gold or ITO contacts were used as electrical contacts at the corners of Hall samples. For the Hall effect measurement, we used 0.6 T magnetic field amplitude with 0.1 Hz frequency. Due to the low frequency, we did not observe the so-called Faraday Induction. After subjecting the sample to a tenfold increase in current, the Hall mobility showed only a 5% change in perovskite and a mere 0.1% change in silicon, which can be attributed to noise. An LED with an emission wavelength of 827 and 617 nm was employed for generating the free carriers in silicon and perovskite thin films. The light soaking intensity-dependent characterization of the charge transport constants was measured up to one sun light power.

The small flat LED illumination is used for CLIMAT setup and makes the implication of CLIMAT more convenient. The CLIMAT module design is shown in Fig. S1.

### Terahertz spectroscopy

The terahertz setup utilized for TRTS (time-resolved terahertz spectroscopy) experiments is built around a femtosecond laser system that delivers pulses with an energy of 7 μJ and a pulse duration of approximately 70 fs at a repetition rate of 150 kHz. The laser system comprises an amplifier (Rega 9050, Coherent), which is seeded by 800 nm pulses generated by a Titan: Sapphire oscillator (Vitara, Coherent) running at 80 MHz. THz pulses were generated through optical rectification of 800 nm laser pulses in a (110)-oriented ZnTe crystal. Following transmission of these terahertz pulses through the sample, their electric field E was measured by electro-optic detection in a second ZnTe crystal. In a separate branch, the laser pulses with a wavelength of 800 nm are directed onto the sample to photo-generate a sheet carrier concentration of $\Delta ns$ of $9.24 \times 10^{11}$–$3.84 \times 10^{12}$ cm$^{-2}$ per pulse in the sample. The frequency-dependent sum of electron and hole mobility spectrum $\mu_\Sigma(f)$ can be obtained from the sheet photo-conductivity $\triangle\sigma_S(f)$[58].

### Silicon sample

We used a B-doped Czochralski-grown silicon sample with a thickness of 150 μm, size $1 \times 1$ cm$^2$, and dark resistivity 42 Ohm cm. We intentionally chose a sample with a large concentration of defects ($N_t > 10^{14}$ cm$^{-3}$) to demonstrate the capability of CLIMAT to detect charge losses.

### Perovskite thin film

Dimethyl sulfoxide (DMSO, anhydrous), dimethylformamide (DMF, anhydrous), isopropanol (IPA, anhydrous), and methyl acetate (MA, anhydrous) were purchased from Sigma-Aldrich. Ethanol (anhydrous) was ordered from VWR Chemicals. Formamidinium iodide (FAI, >99.99%), methylammonium iodide (MAI, >99%), methylammonium chloride (MACl, >99%) were ordered from Dyenamo. Cesium iodide (CsI, 99.999%) was purchased from abcr GmbH. Lead iodide (PbI$_2$, 99.99%), lead bromide (PbBr$_2$, 99.99%), lead chloride (PbCl$_2$, 99.99%) were ordered from TCI.

### Perovskite film preparation

The glass substrates were cleaned by using Mucasol (2% in DI-water), DI-water, acetone, and IPA for 15 min in ultrasonic bath of each steps. The substrates were cleaned with a brusher before cleaning in the ultrasonic bath with Mucasol. After final ultrasonication, the glass substrates were stored in fresh IPA after cleaning.

Then, the glass substrates were treated by UV-Ozone for 15 min before the gold electron evaporation. 80 nm of the Cr was thermally evaporated before 100 nm gold. All the following fabrication procedures were conducted in nitrogen-filled gloveboxes except for the specific descriptions.

In all, 1.4 M of the perovskite chemicals were scaled before dissolving in 1 mL mixed solvent of DMF and DMSO (DMF:DMSO = 3:1 in volume), including 80 mg CsI, 187.8 mg FAI, 269.8 mg PbBr$_2$, and 355.0 mg PbI$_2$. Those were dissolved completely at room temperature. Then, the dissolved perovskite precursor was transferred to another vial with 1.4 M MAPbCl$_3$ (38.9 mg PbCl$_2$ and 9.5 mg MACl), and finally dissolved at 60 °C within 2 h before using. The dissolved 1.8 eV band-gap perovskite precursor was filtered by a 0.20-μm PTFE filter to remove undissolved particles. 100 μL of the perovskite precursor solution was used for the spin-coating on the glass substrates at a spin speed of 5000 rpm for 45 s duration and 1000 rpm s$^{-1}$ acceleration. The 200 μL antisolvent of the methyl acetate was quickly splashed on the perovskite surface at 25 s after starting the spin coating program. The perovskite substrates were annealed at 100 °C for 30 min. Finally, the sample was encapsulated by using the UV curable glue "bluefix®" for the measurement to prevent the influence of atmosphere.

### Tin perovskite film preparation

The FA$_{0.78}$MA$_{0.2}$EDA$_{0.02}$SnI$_3$ perovskite solution with a concentration of 1.2 M was prepared dissolving 447 mg of SnI$_2$ (Sigma-Aldrich, 99.99%), 165 mg of FAI, 33 mg of MAI, and 6.5 mg of EDAI$_2$ in 1 mL of DMF:DMI (6:1 v/v) solvent mixture. The solution was thoroughly mixed by shaking overnight at room temperature and then filtered using a 0.20-μm PTFE filter. Before spin coating, the solution was diluted with 4-(tert-butyl) pyridine (t-BP) at a 2:1 volume ratio. 100 μL of the as prepared solution was spin coated onto a glass substrate at a spin speed of 5000 rpm for 45 s. 120 μL of p-xylene was used as antisolvent and dripped onto the spinning substrate 17 s after the rotation began. The substrates were annealed at 100 °C for 30 min.

### Y6 and PDINN films

The small molecules 2,2′-[[12,13-Bis(2-ethylhexyl)-12,13-dihydro-3,9-diundecylbisthieno[2″,3″:4′,5′]thieno[2′,3′:4,5]pyrrolo[3,2-e:2′,3′-g][2,1,3]benzothiadiazole-2,10-diyl]bis[methylidyne(5,6-difluoro-3-oxo-1H-indene-2,1(3H)-diylidene)]]bis[propanedinitrile] (Y6) and N,N′-Bis{3-[3-(Dimethylamino)propylamino]propyl}perylene-3,4,9,10-tetra-carboxylic diimide (PDINN) were purchased from 1-Material Inc. Copper thiocyanate (CuSCN), diethyl sulfide (DES), chloroform (CHCl$_3$), and methanol were purchased from Sigma-Aldrich. Y6 film with 100 nm thickness was spin coated on a glass substrate at 1600 rpm from 20 mg/mL CHCl$_3$ solution. PDINN film with 50 nm thickness was spin coated at 800 rpm from 10 mg/mL methanol solution.

## Copper(I) thiocyanate thin film preparation

Copper(I) thiocyanate (CuSCN, 99%, Sigma-Aldrich) was dissolved in diethyl sulfide (DES, 98%, Sigma-Aldrich) at a concentration of 0.5 M. After shaking for 1 h at room temperature, the solution was filtered through a 0.45-μm PTFE filter. 200 μL of the solution was then spin-coated dynamically on the glass substrate at 5000 rpm for 30 s. The wet films were then annealed at 100 °C for 10 min, and exposed to a dry air atmosphere (RH < 1.2%) for 5 days.

## CsPbBr₃ single crystals

Cesium lead bromide single crystals were grown by the vertical Bridgman method. The starting materials, CsBr (6N) and PbBr₂ (5N) (purchased from Alfa Aesar) were fused in a sealed quartz ampoule in the stoichiometric ratio. After the synthesis of polycrystalline material, the crystals were grown in a three-zone vertical Bridgman furnace at a speed of 3 mm/h[59]. Once the crystal was grown, it was cut by a wire-cutting machine, and shaped samples were polished with sandpapers and Al₂O₃ powder (grain size of 1 and 0.3 μm) for the final step.

## Single-crystal solution growth MAPbBr3

Single crystals were grown using the inverse temperature crystallization method.1 MABr (99.99% from Greatcell Solar Materials), and PbBr2 (99.999% from Sigma-Aldrich) were used without further purification. For MAPbBr3, a molar ratio of 1.2 to 1 of MABr to PbBr₂ was dissolved in N,NDimethylformamide (99.8% anhydrous from Sigma-Aldrich) for a 1 M solution. Solutions were filtered using a 0.2μm PTFE membrane syringe filter. Crystals precipitated during gradual heating at a rate of 5 °C/day from room temperature to ~70 °C for MAPbBr₃.

## Synthesis of NaBiS₂ NCs

Ten mg NaH (dry, 90%, Merck), 132 mg triphenyl bismuth (99%, Alfa Aesar) and 32 mg sulfur powder (99.5%, Alfa Aesar) were dissolved in 10 mL degassed OLA (70%, Merck) under stirring at room temperature for 15 min. The solution was heated at 80 or 150 °C for 30 min depending on the desired size of the NCs. All the above processes were performed in an Ar-filled glovebox. Later, the whole solution was cooled down to room temperature in a water bath, and mixed with 6 mL hexane (>95%, Merck) and 14 mL OA (90%, Merck). The mixed solution was stirred for at least 2 h to replace the original oleylamine ligands with the more strongly coordinating oleic acid ligands. Next, the solution was equally divided between four centrifuge tubes (7.5 mL for each). In the first purification process, 7.5 mL ethyl acetate (anhydrous, >99.9%, ROMIL) and 7.5 mL acetone (anhydrous, >99.9%, ROMIL) were added into each tube containing the solution and centrifuged at 7000 rpm for 3 min. The precipitated NCs were re-dispersed in 12 mL toluene and equally divided into 2 centrifuge tubes (6 mL for each). In the second purification process, 3 mL ethyl acetate and 3 mL acetonitrile (anhydrous, >99.9%, ROMIL) were added into each tube containing the solution and centrifuged again at 7000 rpm for 3 min. The precipitated NCs were re-dispersed in toluene, and the solution was filtered by a 0.22-μm PTFE syringe filter. The final concentration of the NaBiS₂ NC solution was 20 mg/mL.

## Contacts preparation

To ensure consistent conditions for all studied samples, they were prepared on pre-patterned substrates with four conductive contacts. The patterned contacts were prepared by sputtering conductive Indium tin oxide (ITO) of 100 nm onto the top of glass substrates. In the case of single crystal samples (SC CSPbBr₃, Silicon), small gold dot contacts were evaporated at the corners of the single crystal samples. We considered the effect of contacts on CLIMAT measurements. For our samples, we chose contacts with a low area ($\delta_{contact}/l_{full\ area} = 0.05$); this ensures the Hall voltage and resistivity error is less than 2%[60].

## Reporting summary

Further information on research design is available in the Nature Portfolio Reporting Summary linked to this article.

## Data availability

The data that support the findings of this study are available from the corresponding author upon reasonable request. The data can be accessed by anyone from the following dataset refs. 61,62. We compared CLIMAT with other methods from the following refs. [44,63–76].

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

## Acknowledgements

The authors gratefully acknowledge Klaus Habicht for dedicating resources and access to the Hall setup. The authors gratefully acknowledge the help of Dieter Neher, Safa Shoaee, Norbert Nickel, Rappich Jorg, Thomas Dittrich, Steve Albrecht, Petr Praus, Katharina Fritsch, Marián Betušiak, Jindřich Pipek, Mykola Brynza, Eduard Belas, Jan Franc, Pavel Moravec, Manasi Pranav, Rutger Schlatmann, Lars Korte, and Roman Grill. A.M. acknowledges financial support from the German Science Foundation (DFG) in the framework of the priority program SPP 2196 and funding from the European Union HORIZON-MSCA-2021-PF-01-01 under grant agreement no. 101061809 (HyPerGreen). This work was supported in part: F.L. acknowledges support from funding by the National ScienceAlexander von HumboldtVolkswagen Foundation (NSF), award No. 2043205and Deutsche Forschungsgemeinschaft through the projects Fabulous (project numbers 450968074) Freigeist Program. Y.-T.H. and R.L.Z.H. thank the UK Engineering and Physical Sciences Research Council for funding (no. EP/V014498/2).

## Author contributions

A.M. conceived the idea and CLIMAT method, designed, and performed the main experiments, contributed to Hall device preparation, performed analysis and simulation model, and wrote the manuscript. F.Y. prepared 3CAT perovskite, T.W.G. prepared CuSCN films, C.F. prepared Tin Perovskite film, A.A.-A. prepared 2PACz monolayer, E.S. prepared Y6 films and contacts, F.L. measured and analyzed PLQY, D.K. contributed to the development of Hall setup, Y.-T.H. and R.L.Z.H. designed and prepared NaBiS$_2$, M.A. prepared MAPbBr$_3$, A.K. prepared CsPbBr$_3$, D.F. and V.S. performed and analyzed the TRMC and TRTS measurements, V.S. contributed to CuSCN and Si characterization, A.A. contributed to project financial support. All authors contributed to the paper revision and approved the final manuscript version.

## Funding

## Competing interests

There is no conflict of interest to declare. A.M. is seeking patent protection related to the subject of this manuscript, in particular the CLIMAT method (EP23173681.0).
