## [Peer Review File · Nature Communications]

Resolving Electron and Hole Transport Properties in Semiconductor Materials by Constant Light Induced Magneto TransportREVIEWER COMMENTS

Reviewer #1 (Remarks to the Author):

In this manuscript Musiienko et al. developed a novel technique, Constant Light Induced Magneto Transport (CLIMAT) method, to acquire many important parameters, including electron and hole mobility, lifetime, diffusion coefficient and length, and quasi-Fermi level splitting. Given the high importance of these parameters in optoelectronic applications, such as solar cells, detector and LED etc. By combining light illumination, electrical current, and magnetic field, CLIMAT method can predict the above parameters for solar cell application without making the full device. To confirm the universality of CLIMAT method, they further characterized twelve materials with drift mobilities and lifetimes. This manuscript explores the possibility to advance optoelectronic devices and materials in various technological fields using extended Hall technique combining with light soaking. This method is potentially very useful because it may obtain multiple important parameters which are critical for most optoelectronic applications. It is the first important to validate CLIMAT method, other than universality. In this aspect this manuscript needs to provide convincing evidence. I think this manuscript can be considered for publication if authors can address the following concerns and significantly improve the manuscript:

1. For the calculation of many parameters, it is certainly necessary to make some assumptions and approximations. But most equations are based on constant parameters, which need to be confirmed, because some of the parameters are not really constant, but change with the stimuli or time. Given silicon is well investigated, is it possible to validate the acquired parameters?
2. All kinds of non-standard conduction, such as grain boundaries, surface, ions, and non-uniformities, are simply classified into parasitic conductivity (σ_S) in one equation, it also needs to validate. In addition, the sample includes interfaces and electrodes, should take into account.
3. Relative to Silicon, perovskites will be difficult to apply. Please consider the unique properties and influence: (1) perovskites are confirmed to exhibit mixed electronic-ionic conduction. (2) halide perovskites exhibit excitation dependent carrier lifetime, (3) under constant illumination perovskite exhibit changed defect trapping and carrier lifetime. Therefore, some of the assumptions are not really correct.
4. In terms of the term of lifetime, it is necessary to define the details, because this word is extensively applied for different meanings, for different physical meanings.
5. In terms of generation rate, according to previous publications, strong Auger effect, defect saturation, could occur at the high-density side. Authors should consider the influence.

Reviewer #2 (Remarks to the Author):

The authors have developed a new characterization technique for measuring the electron and hole charge transport properties in different types of semiconductors. After carefully evaluating the work, I think it is closer to the evolution of the measuring method from the light-induced hall technique. From the material and physical aspects, there are indeed no newly synthesized materials, significant device performance improvements and novel physical phenomena. I suggest the transfer of the manuscript to a more specific journal. I also have the following comments which may be helpful for the improvement of the work.

1. In my opinion, when light illuminates a solar cell, not matter the organic or inorganic solar cells, it generates the splitting of the quasi-Fermi energies and the internal field leads to the

flow of charge carriers to the electrodes. On the other hand, the magneto-transport is more concerned with the spin-related magneto-conductance or magneto-resistance. Namely, the change of conductance in a sweeping field. Therefore, I think the title is inappropriate.

2. From the mathematical expressions 1-5 in the manuscript, and those in the supplementary information, I have not seen any magnetic field dependent physical quantities.

3. Another way for the determination of the carrier mobility is to use the Impedance spectroscopic technique, this has not been mentioned/specified with respect to other techniques in the manuscript.

4. I suggest putting a photographic image of the experimental setup together with the schematic drawing in the supplementary information.

5. Although there are many different materials involved in this work, the authors have not really provided details about the fabrication of materials and patterned electrodes for the "light induced hall transport measurements".

6. The solution made hybrid perovskites are known to have the strong ionic transport behavior in the slow time regime. Moreover, they have the light soaking behavior. Both of them may interfere strongly with the charge transport. However, they have not been carefully discussed and distinguished in the manuscript.

7. The charge/spin transport in organic solid films is the hopping transport, which can be quite different by comparing with silicon and perovskite (i.e., drift and diffusive transport). Theoretically, I do not think the different material systems can be easily unified by the model in the manuscript.

Reviewer #3 (Remarks to the Author):

In this study, the authors have introduced the Constant Light Induced Magneto Transport (CLIMAT) method, which provides a comprehensive analysis of electron and hole mobility, lifetime, diffusion coefficient, length, and quasi-Fermi level splitting. Remarkably, the CLIMAT method facilitates the extraction of 14 material parameters simultaneously. The authors have demonstrated the method's versatility by successfully applying it to a diverse range of semiconductor materials.

However, when comparing the CLIMAT method with the previously developed Carrier-Resolved Photo-Hall (CRPH) method by Gunawan et al., no significant advancement seems apparent. Upon closer examination of these two methods, it becomes evident that their measuring principles are quite similar, and both ensure a high degree of measurement accuracy. While the authors assert that CLIMAT can resolve a wider array of properties than CRPH, encompassing the concentration and charge transport characteristics of both holes and electrons, it's worth noting that these parameters can also be calculated from CRPH using hole and electron mobilities.

In light of these observations, I recommend that the authors emphasize the advanced features of the CLIMAT technology, especially in comparison with the CRPH method. By

highlighting the distinguishing aspects of the CLIMAT method and showcasing its superiority over existing techniques, this would merit high IF publication by itself.

REVIEWER COMMENTS

Reviewer #1 (Remarks to the Author):

In this manuscript Musiienko et al. developed a novel technique, Constant Light Induced Magneto Transport (CLIMAT) method, to acquire many important parameters, including electron and hole mobility, lifetime, diffusion coefficient and length, and quasi-Fermi level splitting. Given the high importance of these parameters in optoelectronic applications, such as solar cells, detector and LED etc. By combining light illumination, electrical current, and magnetic field, CLIMAT method can predict the above parameters for solar cell application without making the full device. To confirm the universality of CLIMAT method, they further characterized twelve materials with drift mobilities and lifetimes. This manuscript explores the possibility to advance optoelectronic devices and materials in various technological fields using extended Hall technique combining with light soaking. This method is potentially very useful because it may obtain multiple important parameters which are critical for most optoelectronic applications. It is the first important to validate CLIMAT method, other than universality. In this aspect this manuscript needs to provide convincing evidence. I think this manuscript can be considered for publication if authors can address the following concerns and significantly improve the manuscript:

Reply:

Dear Reviewer, We would like to express our gratitude for your thoughtful and constructive comments on our manuscript titled "Resolving Electron and Hole Transport Properties in Semiconductor Materials by Constant Light Induced Magneto Transport." We appreciate the opportunity to address your concerns and provide further clarification on the points you have raised. Below, we respond to each of your comments and suggestions:

Comment 1

For the calculation of many parameters, it is certainly necessary to make some assumptions and approximations. But most equations are based on constant parameters, which need to be confirmed, because some of the parameters are not really constant, but change with the stimuli or time. Given silicon is well investigated, is it possible to validate the acquired parameters?

Reply 1

We extended the description of the Constant Light Induced Magneto Transport (CLIMAT) method and added the following text to show which parameters are not constant:

Page 7 lines 193-197 : “Note that charge transport parameters τ , μ , σ , RH, L, QFLS, n, p , and η are not constant and change as a function of generation rate as will be shown later in this study. In our model, we assume drift mobility to be independent of the generation rate as confirmed by the saturation of the Hall mobility to the constant value without a further change, particularly increase, of the Hall mobility upon saturation.”

You raise a valid point regarding the need to validate the acquired parameters, especially in the case of well-studied materials like silicon. We agree that it is crucial to validate our results, and we performed additional silicon characterization and presented new results on mobility and lifetime. According to the new ThZ results, the electron mobility in studied silicon samples is 1581 cm²/Vs

compared to 1564 cm²/Vs found by CLIMAT in our manuscript. The Lifetime 0.17 μs measured by trMC is in a good agreement with the lifetime of 1.2 μs found by CLIMAT. Also, QFLS validated by photoluminescence showed 0.58 eV in good agreement with CLIAMT value 0.56 eV. The new data are shown in Supplementary Information (SI) **page 13 lines 241-244**:

“To validate the charge transport properties probed by CLIMAT, we conducted a comprehensive set of additional measurements using alternative characterization techniques. First, we validate lifetime in SI and perovskite, Fig. S12a-b. The electron dirt mobility in silicon is validated by Thz spectroscopy, Fig. S12c.

Figure S1 | Lifetime in (a) silicon and (b) halide perovskite by trMC and trPL measured at 0.6 and 1 sun equivalent. (c) ThZ measurements of mobility in silicon sample.”

Also, new text was added to **the paper page 6 lines 249-250** :

“The value of drift mobility is in good agreement with Terahertz spectroscopy, showing a value of 1581 cm²V⁻¹s⁻¹ (Fig. S12c).”

and **page 9 lines 267-268**:

“The carrier lifetime value aligns well with the results from time-resolved microconductivity, indicating a measurement of 1.7×10⁻⁷ s (Fig. S12a).”

Comment 2

All kinds of non-standard conduction, such as grain boundaries, surface, ions, and non-uniformities, are simply classified into parasitic conductivity (σ_S) in one equation, it also needs to validate. In addition, the sample includes interfaces and electrodes, should take into account.

Reply 2

We added the following text to discuss the effect of contacts on the CLIMAT measurements **page 18 lines 540-542** :

“We considered the effect of contacts on CLIMAT measurements. For our samples, we chose contacts with a low area ($\delta_{\text{contact}}/I_{\text{full area}}=0.05$); this ensures the Hall voltage and resistivity error is less than 2% [Chwang et al.].

[Chwang et al.] Chwang, R., Smith, B. J., & Crowell, C. R. (1974). Contact size effects on the van der Pauw method for resistivity and Hall coefficient measurement. *Solid-State Electronics*, 17(12), 1217–1227. [https://doi.org/10.1016/0038-1101\(74\)90001-X](https://doi.org/10.1016/0038-1101(74)90001-X)”

We compared the value of parasitic conductivity with one reported in the literature and added the following text in the paper on **page 12 lines 327-329**:

“The order of magnitude of σ_s is consistent with previously reported values ($10^{-8} \Omega^{-1}\text{cm}^{-1}$) in halide perovskite semiconductors (Musiienko et al., 2019).”

Musiienko, A., Moravec, P., Grill, R., Praus, P., Vasylchenko, I., Pekarek, J., Tisdale, J., Ridzonova, K., Belas, E., Landová, L., Hu, B., Lukosi, E., & Ahmadi, M. (2019). Deep levels, charge transport and mixed conductivity in organometallic halide perovskites. *Energy & Environmental Science*, 12(4), 1413–1425. <https://doi.org/10.1039/C9EE00311H>

Comment 3

Relative to Silicon, perovskites will be difficult to apply. Please consider the unique properties and influence: (1) perovskites are confirmed to exhibit mixed electronic-ionic conduction. (2) halide perovskites exhibit excitation dependent carrier lifetime, (3) under constant illumination perovskite exhibit changed defect trapping and carrier lifetime. Therefore, some of the assumptions are not really correct.

Reply 3

We carefully consider the mentioned effects raised by the reviewer point by point:

- (1) Mixed conductivity is considered as a part of parasitic conductivity and it involved only a very low generation rate of $10^{19} \text{cm}^{-3}\text{s}^{-1}$. At higher generation rates its impact is negligible (more than 10 x lower, Figure 4a) than the contribution of photogenerated carries to conductivity and Hall effect signal.
- (2) The excitation-dependent carrier lifetime is proven by CLIMAT and shown in Figure 4e.
- (3) Carrier lifetime consists of both recombination, trapping and radiative recombination. We extended the discussion of generation dependence of carrier lifetime in **page 7 lines 185-187**:

“The value of τ probed by CLIMAT accounts for both radiative and non-radiative (trap-associated) recombination processes, as well as takes into consideration trap filling (Eq. S7-8).”

And laso SI **page13 lines 230-234**:

“Considering Eq. S4-6 , the lifetime of electrons and holes is shown in Eq. S7-8. It includes contributions that depend on the generation rate, encompassing both radiative and non-radiative (trap-associated) recombination processes and considers trap filling and occupation saturation.

$$\tau_e = \frac{\Delta n}{G} = \frac{1}{C_{bb}p + \sigma_e v_e (N_t - n_t)} \quad S1$$

$$\tau_h = \frac{\Delta p}{G} = \frac{1}{C_{bb}n + \sigma_h v_h n_t} \quad S2$$

Comment 4

In terms of the term of lifetime, it is necessary to define the details because this word is extensively applied for different meanings, for different physical meanings.

Reply 4

We understand your point regarding the term "lifetime" and its different meanings in various contexts. We extended the definition of charge carrier lifetime **on page 7 lines 184-187**:

“It's worth mentioning that the lifetime, τ , of free carriers is defined as the ratio between photogenerated carrier density and generation rate. The value of τ probed by CLIMAT accounts for both radiative and non-radiative (trap-associated) recombination processes, as well as takes into consideration trap filling (Eq. S7-8).”

Comment 5

5. In terms of generation rate, according to previous publications, strong Auger effect, defect saturation, could occur at the high-density side. Authors should consider the influence.

Reply 5

We considered effect of Auger recombination and added the following text on the SI **page 13 lines 236-238**:

“Auger recombination is neglected, as it only plays a role in carrier recombination at concentrations exceeding (Treglia et al., 2022) 10^{18} cm^{-3} , which is three orders of magnitude higher than the concentrations observed in our study.”

Treglia, A., Ambrosio, F., Martani, S., Folpini, G., Barker, A. J., Albaqami, M. D., de Angelis, F., Poli, I., & Petrozza, A. (2022). Effect of electronic doping and traps on carrier dynamics in tin halide perovskites. *Materials Horizons*, 9(6), 1763–1773. <https://doi.org/10.1039/D2MH00008C>

Reviewer #2 (Remarks to the Author):

Comment 1

The authors have developed a new characterization technique for measuring the electron and hole charge transport properties in different types of semiconductors. After carefully evaluating the work, I think it is closer to the evolution of the measuring method from the light-induced hall technique. From the material and physical aspects, there are indeed no newly synthesized materials, significant device performance improvements and novel physical phenomena. I suggest the transfer of the manuscript to a more specific journal. I also have the following comments which may be helpful for the improvement of the work.

Reply 1

Dear Reviewer, We would like to express our gratitude for your thoughtful and constructive comments on our manuscript. We appreciate the opportunity to address your comments, provide further clarification, and further improve the manuscript. Below, we respond to each of your comments and suggestions:

Comment 2

In my opinion, when light illuminates a solar cell, not matter the organic or inorganic solar cells, it generates the splitting of the quasi-Fermi energies and the internal field leads to the flow of charge carriers to the electrodes. On the other hand, the magnetotransport is more concerned with the spin-related magneto-conductance or magneto-resistance. Namely, the change of conductance in a sweeping field. Therefore, I think the title is inappropriate.

Reply 2

In our study, we have developed a new method called "Constant Light-Induced Magneto Transport" (CLIMAT). The part of this method use the Hall effect, drifting electrons and holes interacts (not spin) with a magnetic field and induce the voltage (Hall voltage V_H) across the sample. In CLIMAT, Light illumination is utilized to control the charge carrier injection level (concentration), which indeed leads to quasi-Fermi level splitting, as demonstrated in the **Figure 2d and Figure 3c** mentioned earlier. Both the concentration of free carriers and quasi-Fermi level splitting are synonymous and connected by the equation $QFLS_h = kT \ln(\frac{p}{n_i})$, given in **page 6 line 183**.

The Hall effect, both with and without light, plays a crucial role in characterizing charge transport in semiconducting materials. We have extended **Table S1 in Supplementary Information (SI)**, highlighting the development of the Hall effect and the photo-Hall method, as well as the resolved charge transport properties. We present this table here for your convenience:

Table S1. Comparison of methods resolving charge transport properties of free carriers based on Hall measurements where σ , R_H , and μ_H are conductivity, Hall coefficient, and Hall mobility.

Method	Mobility	Concentration	Charge transport properties	Parasitic conductivity detection and correction
Hall (1849)3	μ_H	$n = \frac{1}{e(R_H)}$	x	x
Hoshl et al 6 1978	x	n≠p calculated numerically	x	x
Rozenshweig et al. 7 1982	x	n≠p calculated numerically	x	x
Chen and Podzorov et al. 9 2016	$\mu_h \gg \mu_e$	$n = \frac{1}{e(R_H)}$	τ, L, D_e, D_h *	x
Musiienko et al. 10 2018	μ_h	n≠p calculated numerically	x	x
Gunawan et al. 8 2019	$\mu_h = \mu_H^{dark}$ $\Delta\mu = (2 + \frac{d \ln R_H}{d \ln \sigma}) \sigma R_H$	$\Delta n = \Delta p = \frac{\sigma(1 - \mu_e/\mu_h) - e\Delta\mu p_0}{(e\Delta\mu(1 + \mu_e/\mu_h))}$ $\frac{\mu_e}{\mu_h} = \frac{2\sigma(\Delta\mu - \sigma R_H) - e\Delta\mu^2 p_0 \pm \Delta\mu \sqrt{e p_0 \cdot (2\sigma(\Delta\mu - \sigma R_H)) \cdot \sqrt{e\Delta\mu^2 p_0 + 4\sigma(\sigma R_H - \Delta\mu)}}}{(2\sigma(\Delta\mu - \sigma R_H))}$	τ, L, D_e, D_h *	x
Bruevich and Podzorov et al. 67 2021	$\Delta\mu_e = (\mu_h - \mu_{photo-Hall}) \frac{\sigma}{\sigma - \sigma_0}$	$\Delta n = \Delta p = \frac{\sigma - \sigma_0}{(e(\mu_h + \mu_e))}$	τ, L, D_e, D_h *	x
This study 2023	$\begin{cases} p \gg n \Rightarrow \mu_h = \mu_H; \\ n \gg p \Rightarrow \mu_e = \mu_H; \\ p = n \Rightarrow \mu_H = \mu_h - \mu_e = \Delta\mu. \end{cases}$ $\begin{cases} p \gg n \wedge \mu_h > \mu_e \Rightarrow \mu_e = \mu_h - \Delta\mu \\ p \gg n \wedge \mu_h < \mu_e \Rightarrow \mu_e = \mu_h + \Delta\mu \\ n \gg p \wedge \mu_e > \mu_h \Rightarrow \mu_h = \mu_e - \Delta\mu \\ n \gg p \wedge \mu_e < \mu_h \Rightarrow \mu_h = \mu_e + \Delta\mu \end{cases}$	Both electron and hole properties $n = \frac{\sigma \times (\mu_h - R_H \sigma)}{(e(\mu_h \mu_e + \mu_e \mu_e))}$ $p = \frac{(\sigma/e - n \cdot \mu_e)}{\mu_h}$	$\tau_e, \tau_h, L_e, L_h, D_e, D_h, QFLS_c, QFLS_h, \eta$	$\mu_h - \mu_e = \frac{(\mu_{H2}\sigma_2 - \mu_{H1}\sigma_1)}{(\sigma_2 - \sigma_1)}$ $\sigma_S = \sigma_0 - e\mu_h p_0$ $p_0 = \mu_H \sigma_0 / (q_c(\mu_h)^2)$

X – incapable; *Property of only one carrier type

According to the majority of existing literature, Hall effect measurements – the fundamental basis of CLIMAT – are considered one of the magnetic transport phenomena. To clarify what my mean by magnetotransport, we added an explanation in **page 16 lines 424-428**:

“In the context of magnetotransport, we refer to the transport of charge carriers (such as free electrons or free holes) in the presence of a magnetic field, which includes phenomena like the Hall effect. The Hall effect is a part of the magnetotransport measurements family [Dresselhaus *et al.*, 2018; Hamaguchi, 2010; Ziman, 2001]. Hall effect is used to investigate the charge transport properties of materials and as a sensor for measuring the strength of a magnetic field.”

Below, we provide examples from this literature [Dresselhaus *et al.*, 2018; Hamaguchi, 2010; Ziman, 2001] to demonstrate this point. Thus, we believe that the title of the paper is well justified:

Dresselhaus, M., Dresselhaus, G., Cronin, S. B., & Gomes Souza Filho, A. (2018). Magnetotransport Phenomena. 211–230. https://doi.org/10.1007/978-3-662-55922-2_10

Ziman, J. M. (2001). TRANSPORT PHENOMENA IN A MAGNETIC FIELD. *Electrons and Phonons*, 483–524. <https://doi.org/10.1093/ACPROF:OSO/9780198507796.003.0012>

Hamaguchi, C. (2010). Magnetotransport Phenomena. *Basic Semiconductor Physics*, 287–332. https://doi.org/10.1007/978-3-642-03303-2_7

Comment 3

From the mathematical expressions 1-5 in the manuscript, and those in the supplementary information, I have not seen any magnetic field dependent physical quantities.

Reply 3

In Hall effect , Hall effect coefficient, R_H ,depends on magnetic filed B as shown ($R_H = V_{HD}/IB$) in **page 5 line 139** . Hall effect coefficient, R_H , appear in in the Equation 1. We added clarification in main paper **page 5 line 141**:

“Note that R_H depends on magnetic filed B, applied perpendicular to the sample. “

Comment 4

Another way for the determination of the carrier mobility is to use the Impedance spectroscopic technique, this has not been mentioned/specified with respect to other techniques in the manuscript.

Reply4

We are grateful that you suggested mentioning an additional method - The impedance spectroscopic technique. We added it in **SI Table 2, page 3**:

Table S2. Comparison of methods resolving charge transport properties of free carriers: Space Charge Limited Current (SCLC), Drive-level Capacitance Profiling (DLCP), **Impedance Spectroscopy (IS)**, Time-resolved Photoluminescence (trPL), Time-Resolved Microwave Conductivity, Optical-Pump Terahertz-Probe (OPTP), Photoluminescence (PL), carrier-resolved photo-Hall (CRPH), Time of Flight (ToF), and the Constant Light Induced Magneto Transport (CLIMAT).

Method	Steady-state properties	Resolves concentration of holes and electrons	Resolve transport properties holes and electrons	Number of properties	Charge transport properties
DLCP ^{68,69}	⊗	⊗	⊗	1	$N_t(x)$
trPL ⁵¹	⊗	⊗	⊗	1	τ^*
SCLC ⁶⁵⁻⁶⁷	⊗	⊗	⊗	2	N, μ_{\square}^*
IS	⊗	⊗	⊗	2	μ, n or p^*

Comment 5

I suggest putting a photographic image of the experimental setup together with the schematic drawing in the supplementary information.

Reply 5

We added a photographic image of the experimental setup together with the schematic drawing in SI Figure 1 page 1 :

Figure S2 | CLIMAT Photo-Hall Effect measurements module design (a) Schematic drawing of CLIMAT module design; (b) image of the experimental setup.

Comment 6

Although there are many different materials involved in this work, the authors have not really provided details about the fabrication of materials and patterned electrodes for the “light induced hall transport measurements”.

Reply 6

We are grateful for your suggestion. We added the details on fabrication of materials and patterned electrodes in revised paper page 17, lines 536-540:

“To ensure consistent conditions for all studied samples, they were prepared on pre-patterned substrates with four conductive contacts. The patterned contacts were prepared by sputtering conductive Indium tin oxide (ITO) of 100 nm onto the top of glass substrates. In the case of single crystal samples (SC CSPbBr₃, Silicon), small gold dot contacts were evaporated at the corners of the single crystal samples.”

Comment 7

The solution made hybrid perovskites are known to have the strong ionic transport behavior in the slow time regime. Moreover, they have the light soaking behavior. Both of them may interfere strongly with the charge transport. However, they have not been carefully discussed and distinguished in the manuscript.

Reply 7

Mixed conductivity can be considered as a part of parasitic conductivity and it may be involved only at very low generation rates of $10^{19} \text{ cm}^{-3}\text{s}^{-1}$. At higher generation rates its impact is negligible (more than 10 x lower, Figure 4a) than the contribution of photogenerated carriers to conductivity and Hall effect signal. We extended the discussion of and added new data on ionic transport behaviour **SI page 6 lines 112-119**:

“In this study, we do not observe the effect of the ion migration on the measurements due to the very low ion mobility and corresponding low to the electrical field of 10 V cm^{-1} used in the measurements. As a result the long transit time of mobile ions which reaches 10000 s [Cheng, Y, et al. and Musiienko, A et al.] in samples with contact distance of 2 mm which is much larger than typical measurement time of 100 s. To prevent sample degradation on ambient air, samples were encapsulated and stored in a dry glovebox. We measured sample resistivity to control possible sample degradation. We observed a constant resistivity of $2 \times 10^8 \Omega \text{ cm}$ before and after all electrical measurements, confirming the negligible effect of illumination or bias on sample degradation.

Cheng, Y., Liu, X., Guan, Z., Li, M., Zeng, Z., Li, H., Tsang, S., Aberle, A. G., & Lin, F. (2021). Revealing the Degradation and Self-Healing Mechanisms in Perovskite Solar Cells by Sub-Bandgap External Quantum Efficiency Spectroscopy. *Advanced Materials*, 33(3), 2006170. <https://doi.org/10.1002/adma.202006170>

Musiienko, A., Ceratti, D. R., Pipek, J., Brynza, M., Elhadidy, H., Belas, E., Betušiak, M., Delport, G., & Praus, P. (2021). Defects in Hybrid Perovskites: The Secret of Efficient Charge Transport. *Advanced Functional Materials*, 31(48), 2170355. <https://doi.org/10.1002/ADFM.202170355>”

Comment 8

The charge/spin transport in organic solid films is the hopping transport, which can be quite different by comparing with silicon and perovskite (i.e., drift and diffusive transport). Theoretically, I do not think the different material systems can be easily unified by the model in the manuscript.

Reply 8

We agree that charge transport in organic semiconductors can be more complicated than in silicon and perovskite systems due to possible hopping mechanisms. On the other hand, during the jump events of the hopping process, carriers appear to the HOMO or LUMO levels of organic semiconductors, which will generate photoconductivity and Hall effect signals similar to normal semiconductors. Recently, we demonstrated that the morphology and microstructure of the organic semiconductor Y6 have a direct influence on the quality of band-like transport, particularly as detected by photo-Hall measurements (Sağlamkaya et al., 2023). Thus, such properties of free carriers in the HOMO/LUMO levels can be used to distinguish two materials with different transport properties. To emphasize that CLIMAT probes only carriers in the charge transport state, we have added the following text **page 13 line 373**:

"Note that for organic semiconductors, CLIMAT probes properties of carriers in the charge transport state [Sağlamkaya et al.]."

[Sağlamkaya, E., Musiienko, A., Shadabroo, M. S., Sun, B., Chandrabose, S., Shargaieva, O., lo Gerfo M, G., van Hulst, N. F., & Shoae, S. (2023). What is special about Y6; the working mechanism of neat Y6 organic solar cells. *Materials Horizons*.
<https://doi.org/10.1039/D2MH01411D>

Reviewer #3 (Remarks to the Author):

Comment 1

In this study, the authors have introduced the Constant Light Induced Magneto Transport (CLIMAT) method, which provides a comprehensive analysis of electron and hole mobility, lifetime, diffusion coefficient, length, and quasi-Fermi level splitting. Remarkably, the CLIMAT method facilitates the extraction of 14 material parameters simultaneously. The authors have demonstrated the method's versatility by successfully applying it to a diverse range of semiconductor materials.

Reply 1

Dear Reviewer, We would like to express our gratitude for your thoughtful and constructive comments on our manuscript titled "Resolving Electron and Hole Transport Properties in Semiconductor Materials by Constant Light Induced Magneto Transport." We appreciate the opportunity to address your comments and provide further clarification on the points you have raised. Below, we respond to each of your comments and suggestions:

Comment 2

However, when comparing the CLIMAT method with the previously developed Carrier-Resolved Photo-Hall (CRPH) method by Gunawan et al., no significant advancement seems apparent. Upon closer examination of these two methods, it becomes evident that their measuring principles are quite similar, and both ensure a high degree of measurement accuracy. While the authors assert that CLIMAT can resolve a wider array of properties than CRPH, encompassing the concentration and charge transport characteristics of both holes and electrons, it's worth noting that these parameters can also be calculated from CRPH using hole and electron mobilities. In light of these observations, I recommend that the authors emphasize the advanced features of the CLIMAT technology, especially in comparison with the CRPH method. By highlighting the distinguishing aspects of the CLIMAT method and showcasing its superiority over existing techniques, this would merit high IF publication by itself.

Reply 2

We are grateful for your recommendation to include a comparison of CLIMAT method with CRPH method and with other methods.

We included a detailed comparison of the methods based on the Hall effect on Supplementary Information **(SI) page 2-3 Table S2 and Table S1**, showing aspects of the CLIMAT method and showcasing its superiority over existing techniques.

A comparison of CLIMAT with other methods reveals that, at the moment, CLIMAT is the only method capable of separately resolving the concentration, lifetime, quasi-Fermi level splitting, and diffusion length of both holes and electrons. The tables S1 and S2 are shown below for your convenience:

Page3 lines 105-106: "A detailed comparison of CLIMAT, showcasing its superiority over existing state-of-the-art characterization methods, is provided in Tables S1 and S2."

Page 14 line 410: “**Our study represents the pioneering achievement of simultaneously determining all characteristics of minority and majority carriers from a single experimental setup, applied to a single sample, as compared to other methods in Table S1-S2.**”

Table S1. Comparison of methods resolving charge transport properties of free carriers based on Hall measurements where σ , R_H , and μ_H are conductivity, Hall coefficient, and Hall mobility.

Method	Mobility	Concentration	Charge transport properties	Parasitic conductivity detection and correction
Hall (1849)3	μ_H	$n = \frac{1}{e(R_H)}$	x	x
Hoshl et al 6 1978	x	n≠p calculated numerically	x	x
Rozenshweig et al .7 1982	x	n≠p calculated numerically	x	x
Chen and Podzorov et al .9 2016	$\mu_h \gg \mu_e$	$n = \frac{1}{e(R_H)}$	$\tau, L,^*$ D_e, D_h	x
Musiienko et al .10 2018	μ_h	n≠p calculated numerically	x	x
Gunawan et al . 8 2019	$\mu_h = \mu_{H \text{ dark}}$ $\Delta\mu = (2 + \frac{d \ln R_H}{d \ln \sigma}) \sigma R_H$	$\Delta n = \Delta p = \frac{\sigma(1 - \mu_e/\mu_h) - e\Delta\mu p_0}{(e\Delta\mu(1 + \mu_e/\mu_h))}$ $\frac{\mu_e}{\mu_h} = \frac{2\sigma(\Delta\mu - \sigma R_H) - e\Delta\mu^2 p_0 \pm \Delta\mu \sqrt{e p_0} \cdot \sqrt{e\Delta\mu^2 p_0 + 4\sigma(\sigma R_H - \Delta\mu)}}{(2\sigma(\Delta\mu - \sigma R_H))}$	$\tau, L,^*$ D_e, D_h	x
Bruevich and Podzorov et al .67 2021	$\Delta\mu_e = (\mu_h - \mu_{\text{photo-Hall}}) \frac{\sigma}{\sigma - \sigma_0}$	$\Delta n = \Delta p = \frac{\sigma - \sigma_0}{(e(\mu_h + \mu_e))}$	$\tau, L,^*$ D_e, D_h	x
This study 2023	$\begin{cases} p \gg n \Rightarrow \mu_h = \mu_H; \\ n \gg p \Rightarrow \mu_e = \mu_H; \\ p = n \Rightarrow \mu_H = \mu_h - \mu_e = \Delta\mu. \end{cases}$ $\begin{cases} p \gg n \wedge \mu_h > \mu_e \Rightarrow \mu_e = \mu_h - \Delta\mu \\ p \gg n \wedge \mu_h < \mu_e \Rightarrow \mu_e = \mu_h + \Delta\mu \\ n \gg p \wedge \mu_e > \mu_h \Rightarrow \mu_h = \mu_e - \Delta\mu \\ n \gg p \wedge \mu_e < \mu_h \Rightarrow \mu_h = \mu_e + \Delta\mu \end{cases}$	Both electron and hole properties $n = \frac{\sigma \times (\mu_h - R_H \sigma)}{(e(\mu_h \mu_e + \mu_e \mu_e))}$ $p = \frac{(\sigma/e - n \cdot \mu_e)}{\mu_h}$	$\tau_e, \tau_h,$ $L_e, L_h,$ D_e, D_h QFLS _c QFLS _h η	$\frac{\mu_h - \mu_e}{(\mu_{H2} \sigma_2 - \mu_{H1} \sigma_1)} = \frac{(\mu_{H2} \sigma_2 - \mu_{H1} \sigma_1)}{(\sigma_2 - \sigma_1)}$ $\sigma_s = \sigma_0 - e\mu_h p_0$ $p_0 = \mu_H \sigma_0 / (q_c (\mu_h)^2)$

X – incapable; *Property of only one carrier type

Table S2. Comparison of methods resolving charge transport properties of free carriers: Space Charge Limited Current (SCLC), Drive-level Capacitance Profiling (DLCP), **Impedance Spectroscopy (IS)**, Time-resolved Photoluminescence (trPL), Time-Resolved Microwave Conductivity, Optical-Pump Terahertz-Probe (OPTP), Photoluminescence (PL), carrier-resolved photo-Hall (CRPH), Time of Flight (ToF), and the Constant Light Induced Magneto Transport (CLIMAT).

Method	Steady-state properties	Resolves concentration of holes and electrons	Resolve transport properties holes and electrons	Number of properties	Charge transport properties
--------	-------------------------	---	--	----------------------	-----------------------------

DLCP ^{68,69}	☹	☹	☹	1	$N_i(x)$
trPL ⁵⁰	☹	☹	☹	1	τ^*
SCLC ⁷⁰⁻⁷²	☹	☹	☹	2	N_t, μ^*
IS	☹	☹	☹	2	μ, n or p^*
Hall effect ³	☹	☹	☹	2	μ_H, n or p
PL ^{73,74}	😊	☹	☹	2	$\tau, QFLS^*$
Photoconductivity ⁹	😊	☹	☹	3	$n+p, \mu_h + \mu_e$ τ^*
TRMC ⁵⁰	☹	☹	☹	4	$n+p, \mu_h + \mu_e$ τ^*, L^*
OPTP ⁵⁰	☹	☹	☹	4	$n+p, \mu_h + \mu_e$ τ^*, L^*
ToF ^{75,76}	☹	😊/☹- only low injection regime, transient	😊/☹- only low injection regime, transient	4	μ_h, μ_e τ_e, τ_h
CRPH ⁸	😊	☹	☹	7	$\Delta n = \Delta p$ μ_h, μ_e τ^*, L^*, D_e, D_h
Photo-Hall ⁶⁷	😊	☹	☹	7	$\Delta n = \Delta p$ μ_h, μ_e τ^*, L^*, D_e, D_h
CLIMAT This study	😊	😊	😊	14	n, p $\mu_e, \mu_h,$ $\tau_e, \tau_h,$ L_e, L_h D_e, D_h QFLS _e , QFLS _h QFLS, η

😊 -- capable; ☹- incapable; *Property of only one carrier type

REVIEWER COMMENTS

Reviewer #1 (Remarks to the Author):

Authors have carefully considered referees' comments and addressed most the concerns. However, the most important concerns have not yet solved. The CLIMAT method indeed obtains many parameters. The problem is that there exists apparent deviation between the acquired parameters and actual physical meaning of the parameters, under current simplification and assumption. Moreover, as important feature, halide perovskites exhibit significant light soaking effects, either positive or negative, not in negligible level. This means some parameters are the function of the illumination time, in addition to intensity/carrier concentration. I think it is critical to address this problem before it can be considered for publication.

Reviewer #2 (Remarks to the Author):

The authors have almost addressed my concerns, and now I have no further comment on the paper.

Reviewer #3 (Remarks to the Author):

The authors have addressed my concerns, and I suggest this manuscript for publication.

REVIEWER COMMENTS

Reviewer #1 (Remarks to the Author):

Comment 1: Authors have carefully considered referees' comments and addressed most the concerns.

However, the most important concerns have not yet solved. The CLIMAT method indeed obtains many parameters. The problem is that there exists apparent deviation between the acquired parameters and actual physical meaning of the parameters, under current simplification and assumption.

Reply:

We extended definition of lifetime and mobility in page 6 , paragraph 2:

It's worth mentioning that lifetime, τ , of free carriers is defined as ration between photogenerated carrier density and generation rate similar to previous reports[8,11,44]. Steady state τ represents time and L represents distance during which mobile charge carriers (electrons or holes) remain in an active, transport-ready state before undergoing recombination or becoming trapped by defects.

[8]. Gunawan, O. *et al.* Carrier-resolved photo-Hall effect. *Nature* **575**, 151–155 (2019).

[11]. Chen, Y. et al. Extended carrier lifetimes and diffusion in hybrid perovskites revealed by Hall effect and photoconductivity measurements. *Nat Commun* 7, 12253 (2016).

[44]. Bruevich, V., Choi, H. H. & Podzorov, V. The Photo-Hall Effect in High-Mobility Organic Semiconductors. *Adv Funct Mater* 31, 2006178 (2021).

Comment 2:

Moreover, as important feature, halide perovskites exhibit significant light soaking effects, either positive or negative, not in negligible level. This means some parameters are the function of the illumination time, in addition to intensity/carrier concentration. I think it is critical to address this problem before it can be considered for publication.

Reply: We added new data for demonstrating the stability of transport parameters in the supplementary materials page 15-16, paragraph 4:

To illustrate the stability of the CLIMAT signal and transport properties of the examined perovskite sample, we depict the core transport parameters over time at the highest illumination intensity of 1 sun utilized in this study (see Figure S15). As the study was perturbed by light illumination, the semiconductor sample required time to attain steady-state conditions. Once a steady state was achieved, the sample's transport properties remained stable, exhibiting a standard deviation of 5% attributable to noise in materials. Similar behavior was previously observed by us in several material systems 45,76. Note also that the intensity dependence of CLIMAT concentration is in perfect agreement with the theoretical simulation of intensity dependance Figure 4d.

Figure S15| CLIMAT measurements in perovskite sample a) $\mu \times n$ product (b) lifetime of free carriers

Reviewer #2 (Remarks to the Author):

The authors have almost addressed my concerns, and now I have no further comment on the paper.

Reviewer #3 (Remarks to the Author):

The authors have addressed my concerns, and I suggest this manuscript for publication.

REVIEWERS' COMMENTS

Reviewer #1 (Remarks to the Author):

Authors have addressed my concerns. I am happy to recommend publishing this manuscript.

Halide perovskites have been widely confirmed a feature of ultraslow variation under illumination, in timescale of second to hour, very different from conventional semiconductors. This has to be considered for any practical characterization technique. This is partly why carrier lifetime cannot be a standard measurement to evaluate/compare fabrication quality for perovskites. This comment is a suggestion for the future work, certainly beyond the scope of this manuscript, also depending on the advancement in physics of perovskites.